# Collecting mortality data via mobile phone surveys: A non-inferiority randomized trial in Malawi

**Michael Chasukwa**[1,2], **Augustine T. Choko**[3], **Funny Muthema**[1], **Mathero M. Nkhalamba**[4], **Jacob Saikolo**[1], **Malebogo Tlhajoane**[5], **Georges Reniers**[6], **Boniface Dulani**[1,2], **Stéphane Helleringer**[5,7] *

**1** Institute of Public Opinion and Research, Zomba, Malawi, **2** Department of Political and Administrative Studies, University of Malawi, Zomba, Malawi, **3** Malawi-Liverpool Wellcome Trust Clinical Research Programme, Blantyre, Malawi, **4** Department of Psychology, University of Malawi, Zomba, Malawi, **5** Program in Social Research and Public Policy, Division of Social Science, New York University, Abu Dhabi, United Arab Emirates, **6** Department of Population Health, London School of Hygiene and Tropical Medicine, London, United Kingdom, **7** Department of Sociology, New York University, New York, United States of America

* sh199@nyu.edu

**Data Availability Statement:** All data and replication scripts are available at the following link: https://osf.io/seqpu.

## Abstract

Despite the urgent need for timely mortality data in low-income and lower-middle-income countries, mobile phone surveys rarely include questions about recent deaths. Such questions might a) be too sensitive, b) take too long to ask and/or c) generate unreliable data. We assessed the feasibility of mortality data collection using mobile phone surveys in Malawi. We conducted a non-inferiority trial among a random sample of mobile phone users. Participants were allocated to an interview about their recent economic activity or recent deaths in their family. In the group that was asked mortality-related questions, half of the respondents completed an abridged questionnaire, focused on information necessary to calculate recent mortality rates, whereas the other half completed an extended questionnaire that also included questions about symptoms and healthcare. The primary trial outcome was the cooperation rate, i.e., the number of completed interviews divided by the number of mobile subscribers invited to participate. Secondary outcomes included self-reports of negative feelings and stated intentions to participate in future interviews. We called more than 7,000 unique numbers and reached 3,054 mobile subscribers. In total, 1,683 mobile users were invited to participate. The difference in cooperation rates between those asked to complete a mortality-related interview and those asked to answer questions about economic activity was 0.9 percentage points (95% CI = -2.3, 4.1), which satisfied the non-inferiority criterion. The mortality questionnaire was non-inferior to the economic questionnaire on all secondary outcomes. Collecting mortality data required 2 to 4 additional minutes per reported death, depending on the inclusion of questions about symptoms and healthcare. More than half of recent deaths elicited during mobile phone interviews had not been registered with the National Registration Bureau. Including mortality-related questions in mobile phone surveys is feasible. It might help strengthen the surveillance of mortality in countries with deficient

**Funding:** This study was supported by funding from the Eunice Kennedy Shriver National Institute of Child Health and Human Development (R01HD088516 to SH), the National Institute on Aging (R03AG070660 to SH), the Bill and Melinda Gates Foundation (INV-023211 to GR) and New York University-Abu Dhabi. The funders had no role in study design, data collection and analysis, decision to publish and preparation of the manuscript.

**Competing interests:** The authors have declared that no competing interests exist.

civil registration systems. *Registration*: AEA RCT Registry, #0008065 (14 September 2021).

## Introduction

In many low-income and lower-middle-income countries (LLMICs), few deaths are registered with competent administrative authorities in a timely manner [1–3]. In such settings, mortality statistics are only updated every few years, after periodic household surveys or censuses are conducted. The retrospective data these inquiries generate allow estimating mortality levels for periods stretching from a few months to a few years prior to data collection [4], but they do not allow monitoring mortality in near real-time. Household surveys and censuses are also often postponed or canceled during epidemics or other crises, due to elevated risks of disease transmission or heightened safety concerns [5].

In most LLMICs, the data available to track the mortality impact of health crises are thus obtained from more partial and selective data collection systems. The counts of deaths routinely reported during epidemics often only include the deaths that occur among those who have been diagnosed with the disease [6]. Yet, the coverage of testing services is very limited in LLMICs [7]. Some patients may be lost to follow-up after diagnosis, and surveillance systems predominantly record deaths that occur at health facilities [8], even though many deaths occur at home [9,10]. Epidemics also indirectly affect mortality, for example by disrupting health services [11–13]. As a result, reported counts of deaths only include a fraction of the excess mortality caused by an epidemic [14]. This might foster perceptions that global health crises (e.g., COVID-19) have "spared" LLMICs, and it might preclude such countries from effectively advocating for resources required to mitigate the impact of such crises [15].

In the medium to long-term, strengthening civil registration systems, i.e., the administrative apparatus that routinely records vital events, is the main intervention required to address this data gap [16,17]. Achieving universal birth registration and increasing death registration are thus key indicators of progress towards the sustainable development goals (SDGs). In the short term however, interim data sources are needed to better understand how ongoing crises affect population health in LLMICs. In this study, we investigated the use of mobile phone surveys (MPS) as a tool to collect mortality data.

MPS are surveys in which participants are recruited and interviewed entirely by mobile phone [18]. They are increasingly conducted in LLMICs, for example to monitor risk factors for non-communicable diseases [19] or to document fluctuations in economic activity, schooling or healthcare use [20,21]. They present several advantages over other modes of data collection. Because they are implemented remotely, they remove the need for physical interactions between data collectors and participants, and can be sustained during epidemics [22]. They can also be repeated more frequently than household surveys or censuses, since they require less complicated logistics. Finally, owing to the rapid penetration of mobile phones in rural areas of LLMICs, they might allow monitoring mortality trends outside of large cities more conveniently than other surveillance systems.

Several MPS have already included questions about mortality. During the Ebola epidemic in west Africa, a survey in Monrovia (Liberia) asked randomly selected mobile phone users about recent deaths in their households [5]. Since the beginning of the COVID-19 pandemic, MPS surveys in India and Bangladesh have also monitored recent household deaths [23,24]. In particular, a question about recent COVID deaths was included in a large phone survey, which allowed estimating that the death toll related to the COVID-19 pandemic in India might be 6–7 times larger than officially reported [25].

Such examples of mortality-related MPS remain however isolated. There are concerns that questions about recent deaths might be too sensitive to ask during phone interviews, thus prompting high levels of refusals to participate, or to answer specific questions. In some instances, questions about deaths might upset respondents or trigger distress. Whereas such reactions can be addressed when interviews occur in person (e.g., through signs of empathy), they may be more difficult to mitigate remotely. Questions about deaths might also take too long to administer. There are strict recommendations to keep the duration of MPS short [26], and eliciting mortality data about deaths with sufficient detail requires time. Existing mortality-related MPS have only included limited ascertainments of deaths and their circumstances. Finally, MPS interviewers might not be able to probe and cross-check answers provided by respondents as thoroughly as during in-person interviews. The mortality data generated by MPS might thus be of lesser quality.

In a randomized trial conducted among mobile phone users in Malawi, we assessed whether participation and emotional responses differed between mobile users who were asked to complete a mortality-related interview and users asked to complete an interview about their recent economic activity. While this "control" topic might also generate negative reactions from respondents, particularly at times of increased economic stress, it is widely perceived as acceptable to investigate via MPS, and has repeatedly been included in MPS conducted since the beginning of the COVID-19 pandemic. It thus represents a useful benchmark against which to evaluate the feasibility of mortality-related MPS. Our results provide evidence that mortality data collection via MPS is feasible. They suggest that MPS might constitute a useful tool for strengthening the surveillance of mortality during epidemics and other crises affecting population health in countries with deficient civil registration systems.

## Methods

### Ethics statement

The study was approved by the institutional review boards of New York University—Abu Dhabi (HRPP-2021-93) and the University of Malawi (UNIMAREC, P.07/21/76). All participants provided oral verbal consent prior to participating in the study. The protocol was registered prospectively in the American Economic Association's Registry of randomized Controlled Trials (#0008065). Participants in a 3-day pilot provided feedback that informed study procedures. Other members of the study population were not directly involved in the design of the study. Dissemination of study results is ongoing with local stakeholders in Malawi.

### Trial design

We conducted a non-inferiority randomized trial [27], in which we tested whether a mortality-related MPS did not lead to unacceptably worse response patterns than an MPS focused on recent economic activity. Participants in the mortality survey group were asked to complete either an abridged or an extended questionnaire on deaths. The abridged mortality questionnaire only ascertained information required to measure and triangulate recent mortality rates. The extended mortality questionnaire also included questions about symptoms and circumstances of reported deaths.

### Study setting

*T*he trial was conducted in Malawi, a low-income country in southeastern Africa, with a population of approximately 18 million inhabitants. Despite a legal obligation to register vital

events, and the possibility to register events in decentralized offices and locations, few deaths are reported to the National Registration Bureau, i.e., the authority in charge of vital records. As a result, data series based on death registration, typically used to monitor short-term fluctuations in mortality in high-income countries [28], are not available in Malawi [29]. The most recent household surveys that collected mortality data in Malawi occurred between 2015 and 2017, whereas the national census was conducted in 2018.

The coverage of mobile phone networks is extensive in Malawi: in the nationally-representative Afrobarometer survey conducted in November-December 2020, mobile phone service was available in all randomly selected communities located in urban areas, and in 95% of communities located in semi-urban or rural areas. At that time, 56% of all adults in Malawi personally owned a mobile phone, up from 48% in 2014/15. Ownership of mobile phones however remains more limited in Malawi than in other eastern and southern African countries. According to Afrobarometer data, for example, 62% of adults in Mozambique, 75% of adults in Tanzania and more than 90% of adults in countries like Kenya, Botswana or Eswatini personally owned a mobile phone in 2020/21. There are also large disparities in access to mobile phones between socio-economic groups in Malawi. In urban areas, more than 90% of adult men, and close to 75% of adult women reported that they owned a mobile phone (Fig 1). In semi-urban and rural areas, these proportions were 63% and 42%, respectively. Since the beginning of the COVID-19 pandemic, several MPS have been conducted in Malawi [30–32], but they have not reported recent mortality trends.

## Participants

Study participants were mobile phone users aged 18–64 years old at the time of the study. Mobile phone users younger than 18 years old were excluded due to practical difficulties in obtaining parental consent during MPS. Mobile phone users aged 65 years and older were excluded in part due to difficulties in reaching such age groups during MPS. In addition, people aged 65 years and older are typically excluded from household surveys that collect data on mortality. For example, the demographic and health surveys (DHS) only include women aged 15–49 years old, and men aged up to 54 years old [33]. The population-based HIV impact assessments (PHIA) include individuals aged up to 64 years old [34].

We recruited study participants among users of Malawi's two major mobile networks through random digit dialing (RDD). We worked with Sample Solutions, a firm specializing in the provision of RDD samples. Sample Solutions first generated a list of phone numbers at random using the country's numbering scheme. They then matched this list to a global registry of authorized network subscribers, and excluded numbers that could not be located. Finally, a team of 15 interviewers contacted the selected numbers to introduce the study, assess the eligibility of mobile phone users who were reached, and ask for their consent to participate in interviews.

We implemented sampling quotas based on age, gender and regional residence. We formed 18 sampling strata based on these characteristics, and enrollment continued in each stratum until the quota was filled or until progress towards this quota stopped. All interviews were conducted in local languages (Chichewa, Chitumbuka, Chiyao) or in English, depending on mobile users' preferences. If a mobile user did not speak any of these languages, he/she was not included in the trial.

## Randomization

*A*fter interviewers successfully contacted a mobile phone number, they introduced themselves to the user and stated that they were conducting a survey about the impact of the COVID-19

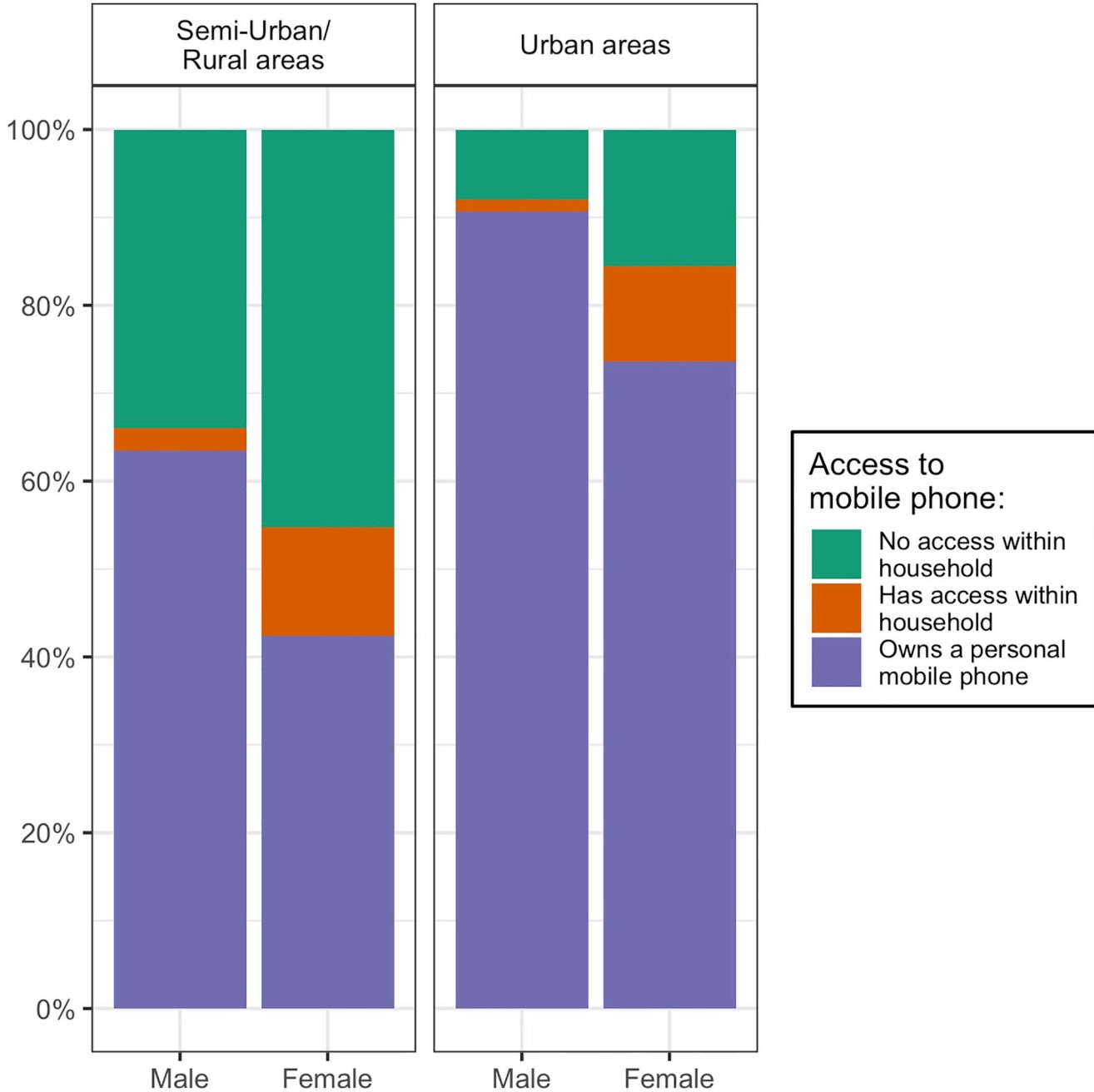

**Fig 1. Reported access to mobile phone in Malawi among adults aged 18 years and older, by gender and area of residence.** Source: Round 8 of the Afrobarometer survey (2020).

pandemic on several aspects of the lives of Malawians. They asked the phone user if they were interested in learning more about the study. If so, they assessed the user's eligibility by asking 4 questions about gender, age and residence. Mobile users who did not meet the age criterion and those whose sampling stratum was already filled were told that they were not eligible for the study. When initially placing a call to a selected mobile number, study interviewers were unaware of the mobile user's assignment to the mortality or economic questionnaire. They

remained unaware of that assignment until the user's sampling stratum had been determined and eligibility had been confirmed.

After learning of the user's assignment to the mortality or economic questionnaire, interviewers read group-specific consent scripts and sought oral consent from mobile users. In both study groups, the consent scripts included similar explanations of how phone numbers were selected, and statements that the interview would take approximately 15 minutes to complete. Consent scripts in both study groups also mentioned that participants would receive a small amount of airtime (1,200 Malawian Kwachas, or 1.5 US Dollar) as a token of appreciation upon completion of the interview. Since the main themes of the interview differed between the mortality and economic questionnaires, the description of other study procedures differed between the two groups.

The randomization process was stratified by sampling stratum. It was conducted using random numbers generated in Stata 15.1. Eligible mobile users were randomized to the mortality or economic questionnaire in a 3:1 ratio. Among phone users interviewed with the mortality questionnaire, randomized allocation to the abridged vs. extended mortality questionnaires was conducted in a similar manner in a 1:1 ratio.

## Outcomes

The primary outcome of the trial was the cooperation rate [35]. It was calculated among mobile users randomized to either the mortality or economic questionnaire, and it was defined as the number of completed interviews divided by the sum of completed interviews, call-backs, partial interviews and refusals. Secondary outcomes included i) the completion rate, i.e. the number of completed interviews divided by the sum of completed interviews and partial interviews and ii) the proportion of respondents who stated their willingness to participate in future interviews of varying durations. We also measured iii) the proportion of respondents who self-reported that some of the survey questions made them upset.

## Sample size

We used standard formulae for the determination of sample size in non-inferiority trials [36]. Based on preliminary pilot data, we assumed a cooperation rate of 86 per cent for both the mortality and economic questionnaires, and we set a non-inferiority margin of 5 percentage points. Our null hypothesis was that the cooperation rate among mobile users asked to complete the mortality questionnaire is lower than among those allocated to the economic questionnaire by more than 5 percentage points. Our alternative hypothesis was that the cooperation rate among those asked to complete the mortality questionnaire is lower by less than 5 percentage points. To test this hypothesis with a 3:1 allocation ration, 80% power and alpha = 0.05, we required at least 1,194 mobile subscribers allocated to the mortality questionnaire and 398 mobile subscribers allocated to the economic questionnaire. Sample size calculations were performed using the SampleSize4ClinicalTrials package in R.

## Study instruments and procedures

In each study group, questionnaires elicited respondents' socio-economic characteristics such as their marital status and educational level, as well as their household's access to water and electricity. In the mortality questionnaire, respondents were asked to list deaths that had occurred among members of their households since the beginning of 2021, and to indicate the survival status of their parents and (maternal) siblings.

Additional questions were included for respondents who had reported that one of their parents or siblings had died since 2019. Respondents allocated to the abridged mortality

questionnaire were solely asked to state the age at death, date of death (month and year) and status of their relative's death in the national civil registration system. Respondents allocated to the extended mortality questionnaire were also asked if their deceased relative(s) had experienced symptoms commonly observed in COVID-19 patients (e.g., cough, fatigue, loss of taste/ smell), and whether they had sought healthcare in the weeks prior to death. Finally, they were asked to indicate the place of death (e.g., at the hospital or at home) and place of burial of their deceased relatives. In the economic questionnaire, instead of questions about deaths, respondents were asked to report their recent economic activities, to list the sources of livelihood of their household, and to describe how they manage their finance (e.g., ownership of bank accounts).

In both study groups, we asked respondents about their reactions to the interview. We adapted debriefing questions used in post-disaster surveys to identify respondents who might have experienced negative feelings during the interview [37]. Participants who self-reported negative feelings were asked to classify these feelings on a 3-point scale ranging from "a little upset" to "very upset". They were also asked if they were still upset by these questions at the end of the interview, or if they were "okay now". Finally, we asked interviewers to indicate if they noticed signs of emotional distress during the interview (e.g., crying, long silences, voice alterations).

We offered psychological support to all respondents who self-reported negative feelings that persisted at the end of the interview, or who were identified by interviewers as having displayed signs of emotional distress during the interview. If a participant indicated being interested in such a service, the study interviewer transmitted the participant's phone number to an on-call clinical psychologist. This practician then called the referred participant to assess their psychological well-being, provide phone-based counseling, and if needed, provide required referrals for further follow-up.

Finally, respondents in both groups were provided details about who to contact if they had questions about the study, and were asked if they wanted to receive information about SARS-- CoV-2. If so, study interviewers read a short script that included details about modes of transmission, symptoms and strategies to prevent infection/transmission. They also indicated how to get additional information through resources provided by the Ministry of Health (e.g., hotline, social media).

### Data collection

Interviewers were trained for six days. Training included a review of recruitment, screening and consent procedures, and explanations of survey questions in each study group. Training also included sessions about the use of tablets for mobile data collection, and the development of skills to conduct of sensitive interviews and detect potential signs of distress among mobile respondents. After mock interviews, interviewers conducted a 3-day pilot. The trial was then conducted between September 21st, 2021 and October 12th, 2021.

All data were collected on tablets using surveyCTO. We recorded how interviewers administered key parts of a randomly selected subset of interviews, including the consent statement, the ascertainment of deaths in the mortality questionnaire, and the assessment of negative reactions to the interview in both study groups. Study supervisors listened to these audio-files to monitor compliance with study instructions, and provided feedback to study interviewers on the basis of these recordings. Study supervisors also placed follow-up calls to a 1 in 15 sample of respondents, which included all participants who experienced negative feelings during the interview, as well as a random sample of respondents who did not experience such reactions. During these follow-up calls, supervisors independently re-assessed participants'

reactions to the interview. They also offered psychological support to participants who indicated experiencing ongoing negative feelings, and gathered other feedback participants had about their interview.

## Statistical methods

We conducted pre-specified intent-to-treat analyses of trial data. We measured primary and secondary outcomes according to participants' assigned study questionnaires, then we computed the differences in those outcomes between mobile users allocated to the mortality and economic questionnaires. We calculated two-sided confidence intervals around these differences [38], and we assessed the position of these confidence intervals relative to the pre-specified non-inferiority margin (-5 percentage points). In computing confidence intervals, we assumed unequal variances in estimates of proportions between the two study groups.

We explored whether respondents with recent deaths among their relatives experienced negative feelings during the interview more frequently than other respondents. We reported the number of participants who asked to obtain counseling and support from our on-call psychologist. In the group of participants allocated to the mortality questionnaire, we described the amount of time required to collect data on mortality during mobile phone interviews. We estimated a linear regression model in which interview duration (in minutes) was the dependent variable, and predictors included a binary variable denoting the type of mortality questionnaire (abridged vs. extended), a categorical variable indicating how many recent deaths were reported by the respondent (none, one or two or more deaths) and an interaction term between those two variables.

We investigated the quality of data on the characteristics of deaths reported during the study. We measured the proportion of reported deaths with missing data on age at death and month of death. We assessed whether these proportions varied by source of the death report (e.g., household deaths, parental deaths or siblings' deaths). Due to small sample sizes in some categories, we used exact methods to calculate confidence intervals [39]. We described the time series of deaths reported during the survey, by source of the death report. Finally, we measured the proportion of reported deaths that were also registered in the national civil registration system, by year of death and source of the death report.

## Results

### Enrollment, participant flow and baseline characteristics

Study interviewers dialed more than 7,000 unique mobile numbers (Fig 2). They reached 3,054 mobile users but 698 users (22.8%) immediately indicated that they were not interested in the study. Only 5 mobile users (0.2%) were excluded due to language-related reasons. Among mobile users whose eligibility was assessed (n = 2,318), 69 did not meet the age-related inclusion criteria (2.9%), and 566 were excluded because their sampling quota had already been filled (24.4%). In total, 1,683 mobile users were randomized to the mortality or economic questionnaire.

Due to preset quotas, the study sample included similar numbers of men and women (Table 1). More than 1 in 4 participants was aged 18–24 years old, whereas 1 in 7 participants was aged 45 years and older. Approximately 20% of the study sample was located in the northern region of Malawi. The southern and central regions each accounted for approximately 40% of the sample. Twenty-eight per cent of the mobile users who were randomized to the mortality or economic questionnaire resided in a city, whereas 49.3% of the mobile users described their place of residence as a town or trading center, and 22.7% were residents of

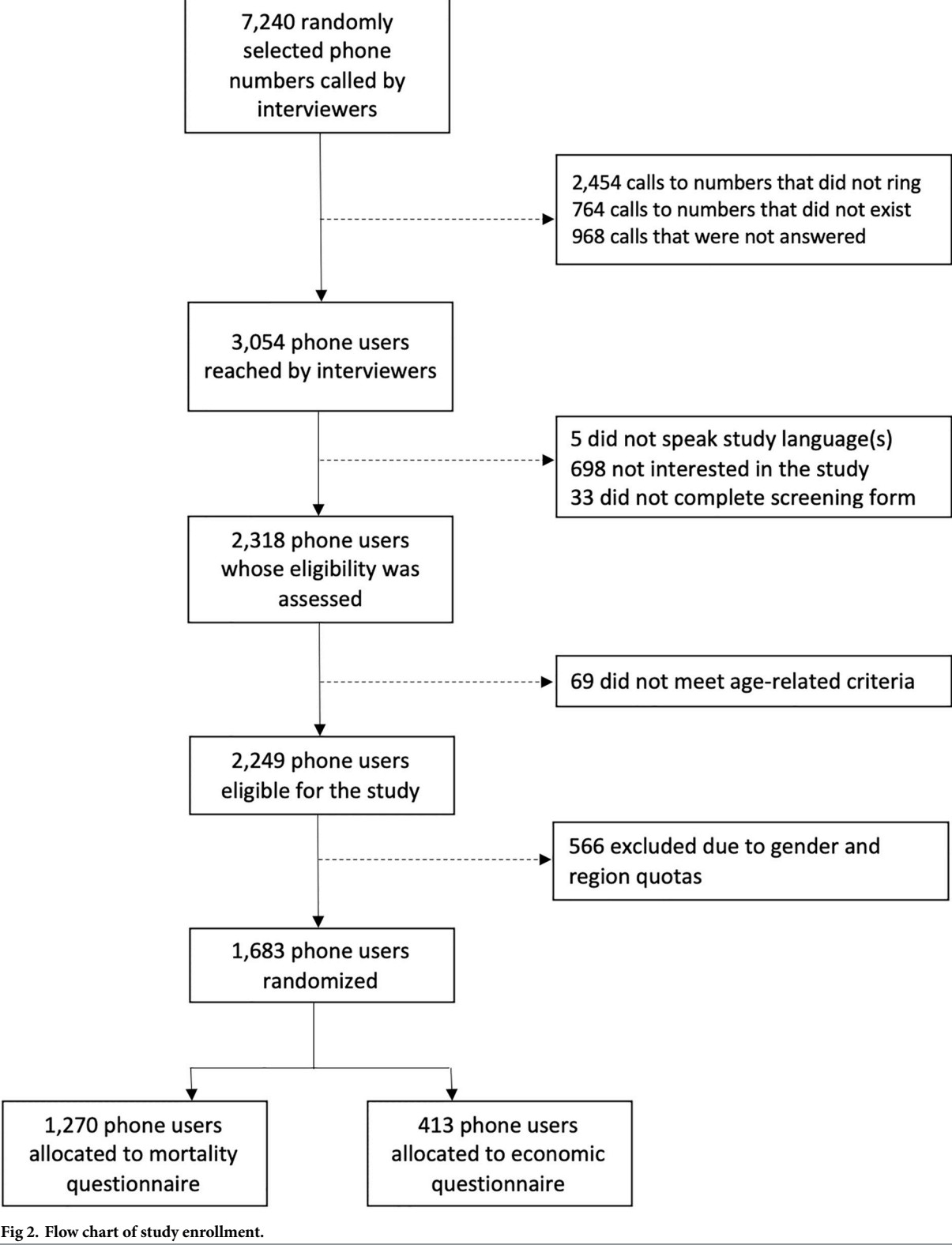

**Fig 2. Flow chart of study enrollment.**

**Table 1. Characteristics of study participants, by study group.**

|  | Mortality questionnaire (N = 1270) | Economic questionnaire (N = 413) | Overall (N = 1683) |
|---|---|---|---|
| **Gender** |  |  |  |
| Male | 650 (51.2%) | 209 (50.6%) | 859 (51.0%) |
| Female | 620 (48.8%) | 204 (49.4%) | 824 (49.0%) |
| **Age group** |  |  |  |
| 18-24y | 317 (25.0%) | 107 (25.9%) | 424 (25.2%) |
| 25-34y | 479 (37.7%) | 170 (41.2%) | 649 (38.6%) |
| 35-44y | 290 (22.8%) | 80 (19.4%) | 370 (22.0%) |
| 45-54y | 134 (10.6%) | 41 (9.9%) | 175 (10.4%) |
| 55-64y | 50 (3.9%) | 15 (3.6%) | 65 (3.9%) |
| **Region** |  |  |  |
| Northern | 260 (20.5%) | 72 (17.4%) | 332 (19.7%) |
| Central | 481 (37.9%) | 180 (43.6%) | 661 (39.3%) |
| Southern | 529 (41.7%) | 161 (39.0%) | 690 (41.0%) |
| **Area of residence** |  |  |  |
| City | 356 (28.0%) | 115 (27.8%) | 471 (28.0%) |
| Town/Trading Centre | 621 (48.9%) | 209 (50.6%) | 830 (49.3%) |
| Villages | 293 (23.1%) | 89 (21.5%) | 382 (22.7%) |

*Notes*: Figures in parentheses are column percentages. The variables included in this table are extracted from the screening form, which was completed by all randomized participants.

villages. There were no important differences in background characteristics between mobile users allocated to the mortality or economic questionnaire.

The distribution of interview results among randomized mobile users is presented in S1 Table. The overall cooperation rate was 92.2% (1,552/1,683). Seventy-eight mobile users refused to provide consent (4.6%). Thirty-two mobile users (1.9%) consented but opted to be called at another time to complete the interview ("call-backs"). Study interviewers could not reach them again despite multiple attempts. Finally, 21 mobile users discontinued their interview (1.2%).

## Primary outcome

*A*mong mobile users allocated to the mortality questionnaire, the cooperation rate was 92.4% (1,174/1,270) vs. 91.5% among users allocated to the economic questionnaire (378/413). This difference of 0.9 percentage points (95% CI = -2.3 to 4.1) meets our criterion for non-inferiority (Fig 3).

## Secondary outcomes

Among those allocated to the mortality questionnaire, 39 out of 1,213 mobile users who consented to participate did not complete their interview (96.8% completion rate, S1 Table) vs. 14 out of 392 mobile users allocated to the economic questionnaire (96.4% completion rate). Similarly, 1,134 of the 1,174 respondents who completed the mortality survey did not self-report that they were upset by some of the interview questions (96.6%) vs. 367 out of 378 among those who completed the economic survey (97.1%). Finally, 94.7% of participants who answered the mortality questionnaire and 93.7% of those who answered the economic questionnaire stated their intention to participate again in (hypothetical) similar interviews in the

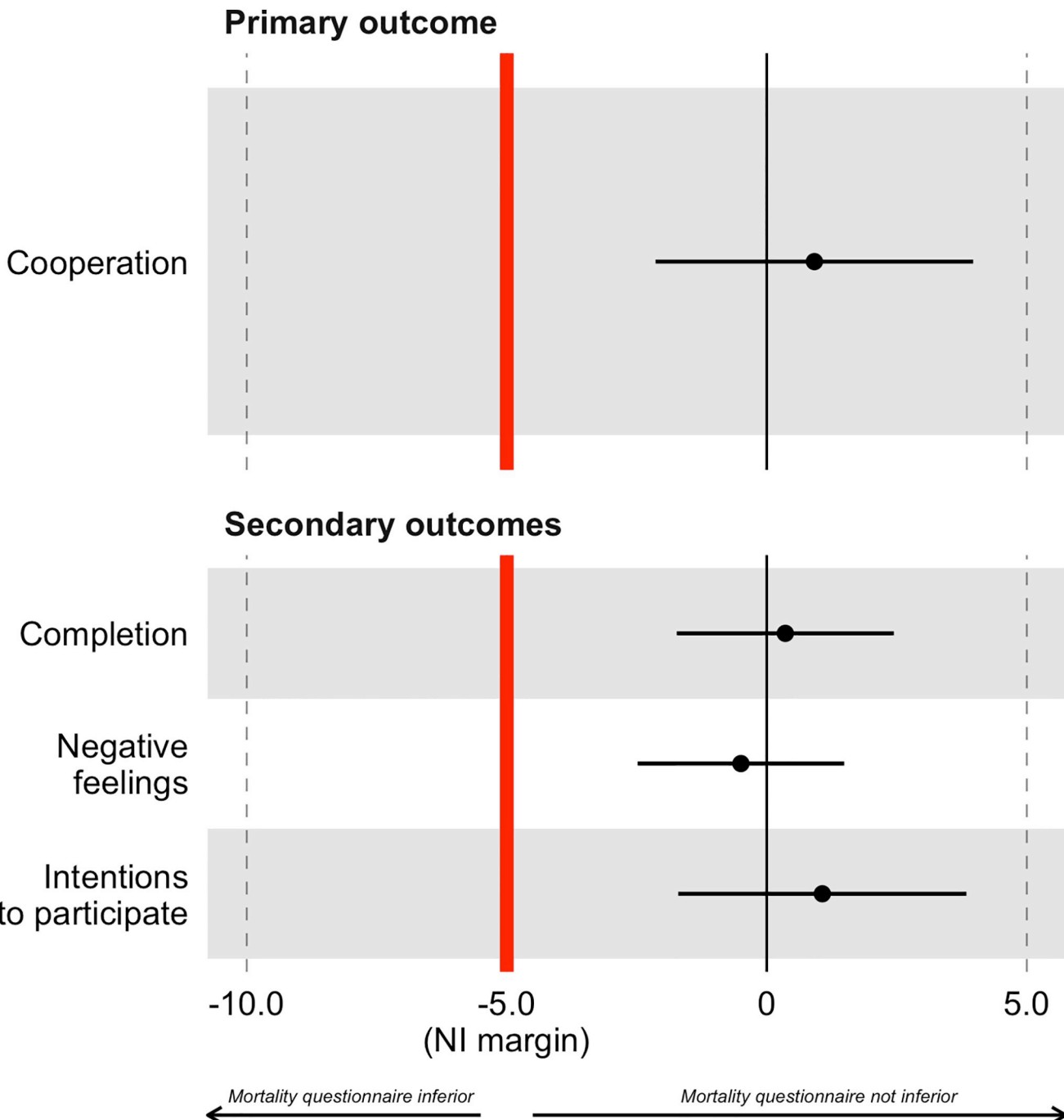

**Fig 3. Differences in study outcomes between mortality and economic questionnaires.** Notes: Values of the x-axis are expressed in percentage points. They are calculated as % in the group allocated to the mortality questionnaire, minus the % in the group allocated to the economic questionnaire. Error bars represent two-sided 95% confidence intervals around the difference in proportions between study groups. The non-inferiority criterion is met when the confidence interval remains to the right of the non-inferiority margin (red vertical line). Similar results were obtained when calculating one-sided confidence intervals. NI = Non-inferiority.

### Referrals to psychological counselling

Fifty-one participants self-reported being upset by (some of) the interview questions. Among 40 participants in the mortality survey group who self-reported negative feelings, 12 indicated that they were "very upset" by some of the interview questions. Participants with a recent death among their relatives reported being "very upset" more frequently than other participants (S1 Fig). Among those who completed the economic survey, all participants who self-reported negative feelings (n = 11) indicated being either "a little upset" or "moderately upset".

Among respondents who self-reported negative feelings, 3 stated that they were still upset at the end of the interview (all in the group allocated to the mortality questionnaire). Study interviewers identified 6 additional participants who displayed signs of distress during the interview, but who did not self-report being upset (3 participants who completed the mortality survey, and 3 who completed the economic survey). In follow-up calls with study supervisors that took place 1–2 days after the interview, a few respondents who did not initially self-report negative feelings indicated that some of the questions asked during the interview had made them upset (S2 Table, n = 7). None of these respondents however reported that these negative feelings persisted at the time of their call with study supervisors. In total, 9 respondents were informed about the possibility to talk to an on-call clinical psychologist who would provide support and information. Three participants (2 in the group allocated to the mortality survey, and 1 in the group allocated to the economic survey) accepted this offer and were contacted. After the call with the clinical psychologist, none of those 3 participants required additional referrals and follow-up.

### Interview duration

On average, study participants allocated to the mortality questionnaire required 18 minutes to complete all study procedures, including screening, consent and debriefing (S2 Fig). Participants who did not report any recent death required on average 16 minutes to complete study procedures (Fig 4). Among participants allocated to the abridged mortality questionnaire, those who reported one recent death among their household members or relatives required an additional 1.9 minute to complete the interview, whereas those who reported 2 or more such deaths required 4.4 additional minutes. Among those who were assigned to the extended mortality questionnaire, similar estimates of additional time required were 4.0 minutes and 12.3 minutes, respectively.

### Data quality

The characteristics of recent deaths reported by mobile respondents are described in Fig 5. The likelihood of missing data on age at death and month of death varied by source of death report, with lowest levels observed among household deaths (0–3%) and highest levels observed among deaths of siblings (>10%, panel A). Counts of reported siblings' deaths varied between 2 and 12 per quarter over the 3 years preceding the survey, whereas reported parental deaths varied between 4 and 14 per quarter (Fig 5, panel B). Questions on household deaths generated between 9 and 21 reports of deaths per quarter in 2021. Between half and two thirds of recent deaths reported during the survey were not registered with the national registration bureau, depending on the source of the report and the year of death (Fig 5, panel C).

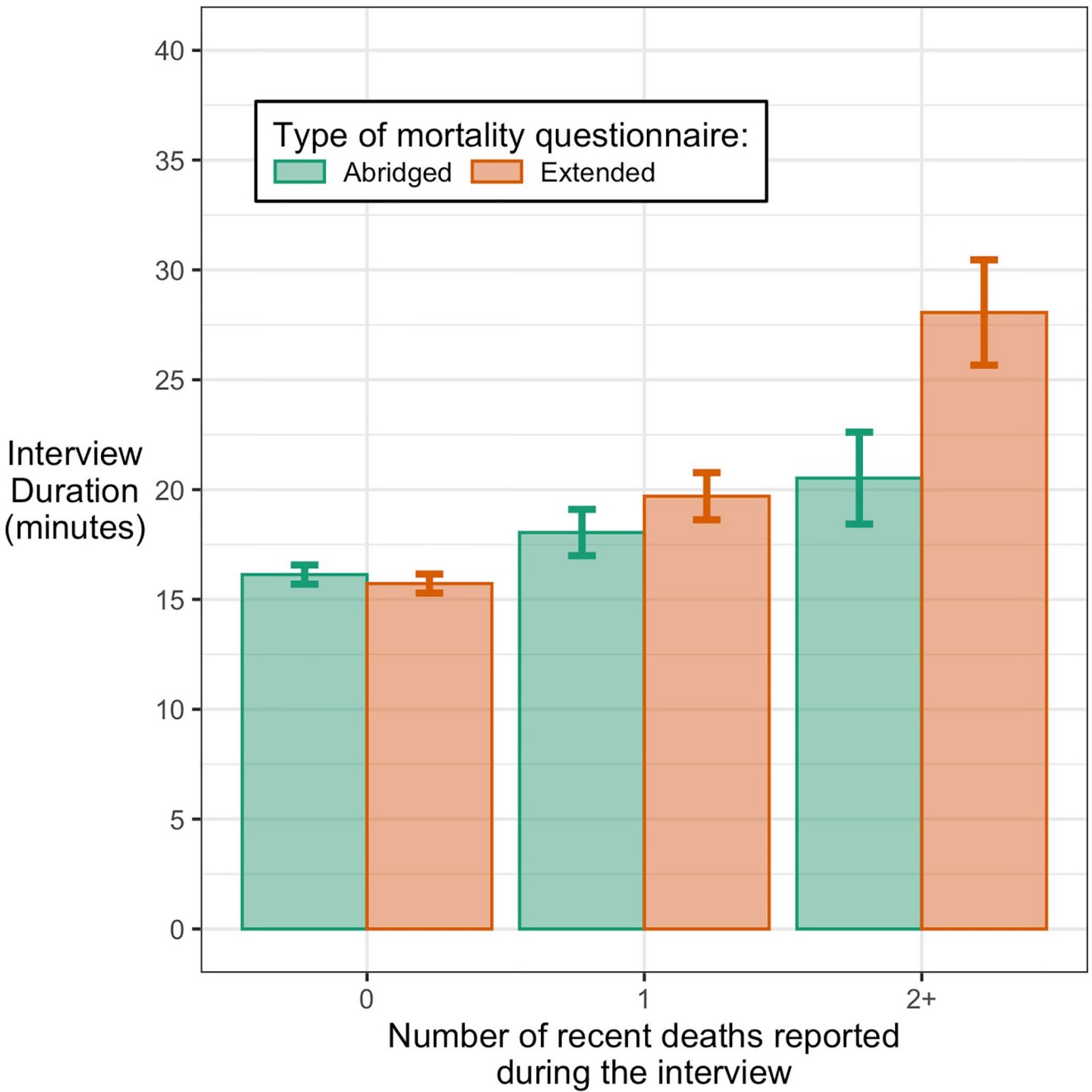

**Fig 4. Duration of treatment group interviews, by type of mortality questionnaire and recent deaths among respondents' relatives.** *Notes*: Error bars represent 95% confidence intervals obtained from linear regression models.

## Discussion

In this randomized trial in Malawi, a mortality-related questionnaire administered by mobile phone was non-inferior to a questionnaire about economic activity recognized as highly acceptable. Cooperation and completion rates were high among respondents asked to answer questions about recent deaths in their households and families. The frequency of self-reported negative feelings was low (<3%), and it was not heightened compared to respondents who

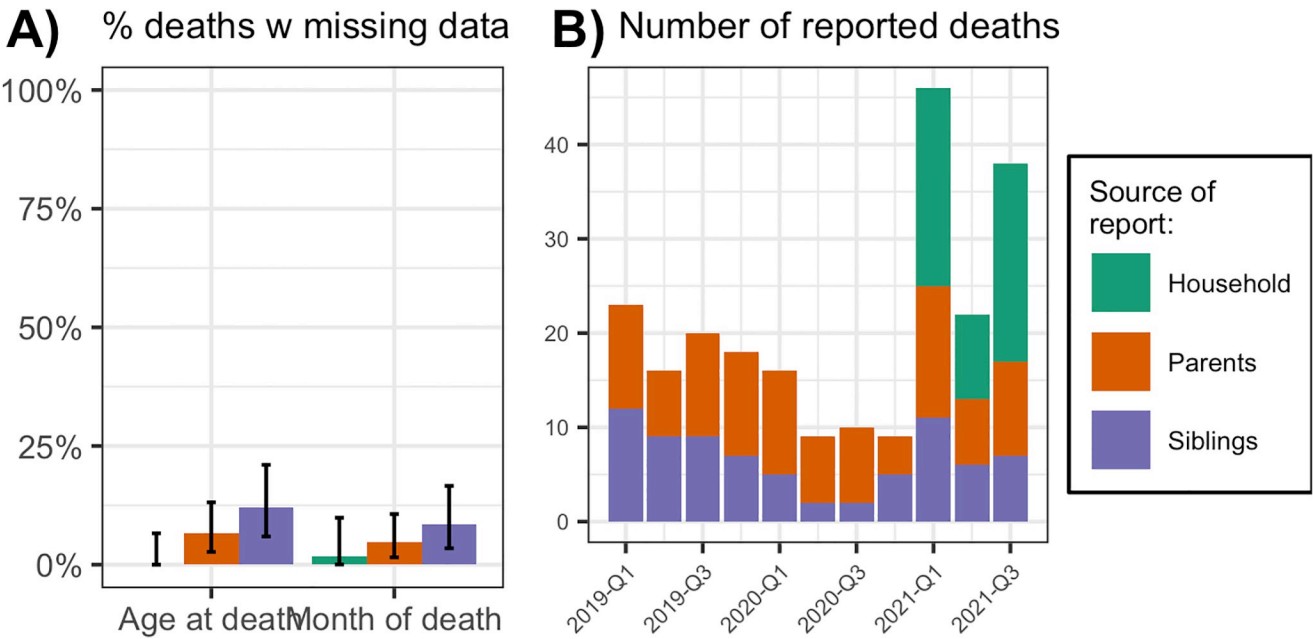

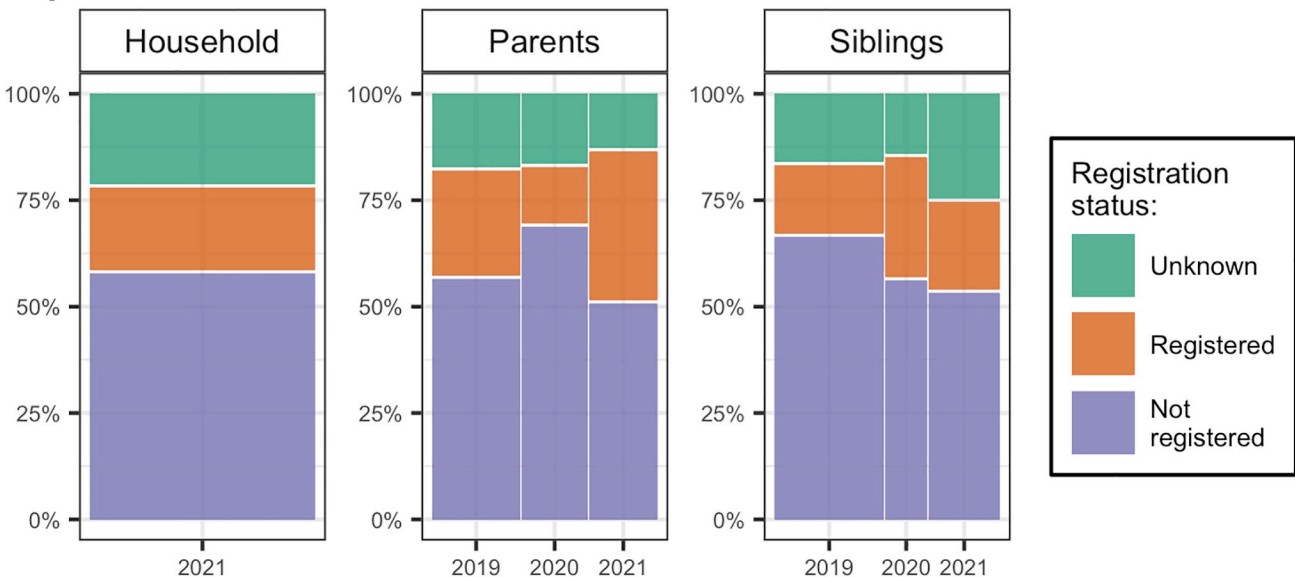

**Fig 5. Exploratory assessments of the quality of reported data on recent deaths.** *Notes*: In Panel A), error bars represent 95% exact confidence intervals. The difference in the likelihood of missing data on age at death by source of report was significant at the p<0.05 level. In Panel B), Q1 refers to the first quarter of a year, i.e., January, February and March. Two deaths reported to have occurred in October 2021 by respondents interviewed after October 1st, 2021 were omitted from the plot. Household deaths were only elicited for the period between the start of 2021 and the survey data. In Panel C), the width of each bar is proportional to the number of deaths reported in each year.

were asked economic questions. Respondents who reported a recent family or household death during the interview experienced stronger negative feelings than other respondents, but these feelings were often transitory, i.e., respondents reported that they had dissipated by the end of the interview. The few respondents who opted to talk to an on-call clinical psychologist

did not require further follow-up. In this trial, concerns that questions related to mortality might be too sensitive to ask during mobile phone surveys were thus not realized.

We used questionnaires that were more detailed than instruments included in previous mortality-related mobile phone surveys, which focused on specific types of deaths (e.g., Ebola or COVID-19 deaths), or only ascertained deaths among members of the respondent's household. We included questions about mortality that emulated those asked during household surveys and censuses. In addition to recent household deaths, they covered survival of close relatives (e.g., siblings, parents) and elicited additional information required to better understand the determinants and context of excess mortality. Asking such questions about mortality required an additional 2–4 minutes per reported death, depending on the inclusion of questions about circumstances, symptoms and healthcare use around the time of death. This time investment concerns only a small group of survey participants, since >80% of respondents did not report any recent death in our mobile phone survey.

Mortality-related phone interviews generated data that appeared of good quality. Missing data on age at death, and date of death, were limited. They were more frequent in reported deaths of siblings, and least frequent in reported deaths of household members. The levels of missing data we observed were comparable to those observed in many household surveys conducted in-person [40]. In this sample, the mortality questions generated reports of approximately 8.4 deaths per quarter over the past 3 years, with marked increases in the most recent time period (i.e., the first 3 quarters of 2021) when questions about household deaths were applicable. Future work should investigate the sample sizes required in mobile phone surveys to allow detecting short-term fluctuations in mortality.

Our data indicate that 50–65% of the deaths that were reported during mobile phone interviews had not been previously registered with the National Registration Bureau, i.e., the administrative unit in charge of civil registration in Malawi. Conducting mortality-related mobile phone surveys might thus supplement existing administrative data collection systems. Mobile phone surveys might also fill gaps in other mortality surveillance systems recommended by the World Health Organization to monitor the death toll of the COVID-19 pandemic and other health crises [41]. These systems entail, for example, extracting data about deaths from registers maintained by health facilities, or tallying the number of burials taking place at local cemeteries. New data sources (e.g., satellite images, social media) might also help keep track of increasing numbers of deaths [42,43].

Unfortunately, setting up such facility-based or community-based surveillance of mortality is complex, particularly during an epidemic or a conflict. It requires significant investments in data acquisition and management. Baseline data that pre-date health crises might be difficult to obtain, for example if historical records were not properly kept at a health facility or cemetery. Behaviors related to the management of deaths (e.g., where and when to carry out a burial) might also change during epidemics and other crises, due to restrictions on mobility or access to health facilities. In such contexts, short-term fluctuations in mortality documented by newly established surveillance systems might be due to behavioral changes as well as excess mortality. Finally, rapid mortality surveillance systems have been predominantly established in large urban areas [44]. Small towns and rural areas are less often included in such initiatives.

By contrast with recommended facility-based and community-based approaches to mortality surveillance, mobile phone surveys are significantly less complex to initiate. Random samples of mobile subscribers can be quickly generated, by using the country's numbering scheme and/or forming partnerships with mobile providers or global sampling platforms. Interviewers can often be trained remotely [45] and work from their own homes [22,31]. By using questionnaires similar to those used in prior household surveys and censuses, the data generated by mobile phone surveys might be compared with previous mortality estimates [23,25]. Finally,

due to the growing penetration of mobile phones in more remote areas, mobile phone surveys have the potential to help assess mortality trends in areas seldom covered by other rapid surveillance systems (e.g., rural or semi-urban areas).

## Our trial has several limitations

First, our sample size was too small to assess the quality of mortality-related data generated by MPS in more detail. We thus could not measure heaping in reported ages of siblings [46], nor could we investigate the reporting of deaths on shorter time scales (e.g., weeks). Second, some of the outcomes we considered (e.g., negative feelings, death registration) were based on self-reported data. They might have been affected by social desirability biases. If the extent of such biases did not differ by trial group, our assessments of the non-inferiority of the mortality questionnaire are however unaffected. In addition, we implemented robustness checks (e.g., audio-recordings, supervisor follow-ups) to enhance the reliability of these data. Finally, we did not explore the acceptability of more extensive mortality questionnaires that might enable the attribution of causes of deaths [47], nor did we investigate the measurement of child mortality through the collection of birth or pregnancy histories [48].

The development of mobile phone surveys as a reliable tool for mortality surveillance in LLMICs requires investigating the selectivity of samples recruited by mobile phone. These samples exclude population members who do not have access to a mobile phone. They disproportionately include younger and more urban respondents [49]. In LLMIC settings, samples of mobile subscribers often predominantly include males [50], who are more likely to be phone owners. Patterns of phone ownership and utilization might also change over the course of an epidemic, for example if poor households are forced to sell mobile phones due to hardship, or are less able to afford charging their phone (and may not be reachable). Mobile users in the rural areas may not be reachable because more limited access to electricity might prevent them from charging their mobile phone consistently. The biases stemming from limited access to mobile phones in local populations might be more pronounced in LLMICs like Malawi where a large proportion of the population does not personally own a mobile phone, and where access to mobile phones varies sharply between socio-economic groups (Fig 1). Biases related to sample selectivity might be more limited in other LLMICs where mobile phone ownership is more widespread (e.g., Kenya). Statistical models that account for such complex patterns of selectivity in estimating mortality rates from survey data collected by mobile phone should be developed and tested.

Despite limitations and ongoing research needs, our work in Malawi suggests that mobile phone surveys are a potentially useful tool for mortality surveillance in LLMICs with limited civil registration systems.

## Supporting information

**S1 Checklist. CONSORT checklist.**
(DOC)

**S1 Fig. Strength of self-reported negative feelings experienced during the interview, by reports of recent deaths and study group.** *Notes*: Respondents were first asked if any of the interview questions made them upset, and if so, they were asked to indicate the severity of their negative feelings. In the group asked mortality-related questions, the strength of negative feelings was associated with recent reports of deaths (p = 0.02). In the mortality-related group group, the width of each bar is proportional to the number of respondents reporting different

numbers of deaths during the interview.
(PDF)

**S2 Fig. Interview duration in the group asked mortality-related questions, by type of mortality questionnaire administered.** Notes: The dashed vertical line represent the mean duration of interviews in the group asked questions related to mortality, regardless of questionnaire type (abridged vs. extended).
(PDF)

**S1 Table. Distribution of study results, by study group.** Notes: "Busy/call-back" refers to respondents who consented to being interviewed, indicated that they would prefer being called-back at a later time, and could not be reached again before the completion of the study.
(DOCX)

**S2 Table. Concordance of self-reported data on negative feeling.** Notes: Supervisor follow-ups were conducted 1–2 days after the initial interview. The supervisors were not aware of the answers provided about self-reported feelings during the initial interview; nor were they aware of the respondents' assignment to the different study groups.
(DOCX)

## Author Contributions

**Conceptualization:** Augustine T. Choko, Boniface Dulani, Stéphane Helleringer.

**Data curation:** Michael Chasukwa, Augustine T. Choko, Jacob Saikolo, Malebogo Tlhajoane, Boniface Dulani, Stéphane Helleringer.

**Formal analysis:** Augustine T. Choko, Jacob Saikolo, Stéphane Helleringer.

**Funding acquisition:** Georges Reniers, Stéphane Helleringer.

**Investigation:** Michael Chasukwa, Mathero M. Nkhalamba, Jacob Saikolo.

**Methodology:** Michael Chasukwa, Augustine T. Choko, Georges Reniers, Stéphane Helleringer.

**Project administration:** Michael Chasukwa, Funny Muthema, Jacob Saikolo, Boniface Dulani.

**Resources:** Michael Chasukwa, Funny Muthema.

**Software:** Jacob Saikolo.

**Supervision:** Michael Chasukwa, Funny Muthema.

**Validation:** Michael Chasukwa, Funny Muthema.

**Visualization:** Augustine T. Choko, Stéphane Helleringer.

**Writing – original draft:** Michael Chasukwa, Stéphane Helleringer.

**Writing – review & editing:** Augustine T. Choko, Funny Muthema, Mathero M. Nkhalamba, Jacob Saikolo, Malebogo Tlhajoane, Georges Reniers, Boniface Dulani.

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
