## [Decision Letter · Decision Letter 0]

20 May 2022

PGPH-D-22-00492

Collecting mortality data via mobile phone surveys: a non-inferiority randomized trial in Malawi

Dear Dr. Helleringer,

Thank you for submitting your manuscript to PLOS Global Public Health. After careful consideration, we feel that it has merit but does not fully meet PLOS Global Public Health’s publication criteria as it currently stands. Therefore, we invite you to submit a revised version of the manuscript that addresses the points raised during the review process.

Please submit your revised manuscript by . If you will need more time than this to complete your revisions, please reply to this message or contact the journal office at globalpubhealth@plos.org. Please include the following items when submitting your revised manuscript:

We look forward to receiving your revised manuscript.

Kind regards,

Rachel Hall-Clifford

Academic Editor

Journal Requirements:

State what role the funders took in the study. If the funders had no role in your study, please state: “The funders had no role in study design, data collection and analysis, decision to publish, or preparation of the manuscript.”

3. Please update your Competing Interests statement. If you have no competing interests to declare, please state: “The authors have declared that no competing interests exist.”

4. In the online submission form, you indicated that your data will be submitted to a repository upon acceptance.  We strongly recommend all authors deposit their data before acceptance, as the process can be lengthy and hold up publication timelines. Please note that, though access restrictions are acceptable now, your entire data will need to be made freely accessible if your manuscript is accepted for publication. This policy applies to all data except where public deposition would breach compliance with the protocol approved by your research ethics board. If you are unable to adhere to our open data policy, please kindly revise your statement to explain your reasoning and we will seek the editor's input on an exemption. Please be assured that, once you have provided your new statement, the assessment of your exemption will not hold up the peer review process.

5. Please provide separate figure files in .tif or .eps format only and ensure that all files are under our size limit of 10MB.

6. CONSORT 2010 Checklist - 01032022sh.doc is currently uploaded as file type “Other”, which is not viewable by the reviewers. Please change the file type(s) to 'Supporting Information' and include a legend in the manuscript if you wish it/them to be included in review. 

7. Please provide an Author Summary. This should appear in your manuscript between the Abstract (if applicable) and the Introduction, and should be 150–200 words long. The aim should be to make your findings accessible to a wide audience that includes both scientists and non-scientists. Sample summaries can be found on our website under Submission Guidelines: https://journals.plos.org/globalpublichealth/s/submission-guidelines#loc-parts-of-a-submission

Alternative link: http://journals.plos.org/ploscompbiol/s/submission-guidelines#loc-author-summary

8. We have noticed that you have uploaded Supporting Information files, but you have not included a list of legends. Please add a full list of legends for your Supporting Information files after the references list.

Additional Editor Comments (if provided):

Thank you for your submission of this exciting work. With the minor revisions suggested by the reviewers, I believe this manuscript will make an important contribution. In particular, the contextualization of rates of mobile phone ownership (and types of access, data, etc.) is highly relevant for explaining the rationale behind the study design and central question. Please also attend to the question regarding sample size calculation.

Reviewers' comments:

Reviewer's Responses to Questions

**Comments to the Author**

1. Does this manuscript meet PLOS Global Public Health’s publication criteria? Is the manuscript technically sound, and do the data support the conclusions? The manuscript must describe methodologically and ethically rigorous research with conclusions that are appropriately drawn based on the data presented.

Reviewer #1: Yes

Reviewer #2: Yes

2. Has the statistical analysis been performed appropriately and rigorously?

Reviewer #1: Yes

Reviewer #2: Yes

3. Have the authors made all data underlying the findings in their manuscript fully available (please refer to the Data Availability Statement at the start of the manuscript PDF file)?

Reviewer #1: Yes

Reviewer #2: Yes

4. Is the manuscript presented in an intelligible fashion and written in standard English?

Reviewer #1: Yes

Reviewer #2: Yes

5. Review Comments to the Author

Reviewer #1: The authors report a randomized trial to assess the acceptability of a mobile phone survey to enhance real-time mortality surveillance in Malawi. This is noble and important work and I applaud the authors for pursuing this line of investigation. The paper is clearly written, and the findings are accurately described and contextualized. I think this article will be of interest to public health officials and NGOs who collaborate in similar settings.

Despite the article’s strengths, there are several weaknesses that I think could be addressed to strengthen the manuscript. My biggest question is why a survey about economics in a low-income country was determined to be an appropriate comparator for acceptability of a mobile phone survey. Would the authors imagine that asking about finances during a global pandemic with economic repercussions in a low-income country may be perceived as intrusive in any way? Yes, the authors mention that this economic survey is “highly acceptable”, but I think it is worth considering and discussing a bit more. I do not think this takes away from the value added of this study, but I do think it is worth considering.

Also, can the authors provide any estimates on phone ownership in Malawi? This may help contextualize the limitation of this method of surveillance being limited to phone owners. I would imagine that families that do not own phones may be more economically disadvantaged and perhaps have higher mortality rates. Otherwise, I have minor comments below for the authors to consider.

Abstract:

-Perhaps it would help the reader understand the primary outcome better if “cooperation rate” were defined in the abstract. This is not a commonly used or widely understood term. Just this just mean that respondents filled it out?

-I think it would help the reader understand the size of the study by including the number of phone numbers that were sent the survey and how many respondents there were in the Abstract.

Introduction:

-Minor, but line 58 should be “and” not “an”.

-In general, this reads quite long and may benefit from some tightening up of the language, though the content is good and sets the stage for the study well. Much of the content can be moved to the Discussion, in my opinion.

-Minor, but please be consistent with the use of COVID-19. Currently there are some areas where it reads COVID and others that read COVID-19.

-The final paragraph, which is currently a sentence, does not clearly state the rationale or potential implications of this line of work. Though this is stylistic, I think this would help the authors succinctly frame the need and the potential implication of this work in a paragraph that many readers look for this information.

Methods:

-In the Randomization section it is unclear if the interviewers knew which group the potential participant would potentially pertain to at the time of the phone call. Was the randomization done prior to the phone call? During the phone call? This could be clearer.

Results:

-Table 1: I’m not sure “treatment” is the appropriate word since this is a survey. Perhaps clarifying the language by stating “Mortality Survey Respondents” or something to that effect would be more appropriate and “Economic Survey Respondents” instead of Control.

Reviewer #2: Reviewer comments

PGPH-D-22-00492 - Collecting mortality data via mobile phone surveys: a non-inferiority randomized trial in Malawi

Thank you for providing me an opportunity to review your research. This is a welcome addition to the field of evidence for the use of Mobile phones for quick surveys of mortality.

I must congratulate on getting out a fairly well written without grammatical errors or spelling mistakes. I have provided some comments within the PDF attached - some of them are suggestions or observations on your text with the hope that you may modify or provide justifications for not doing so.

One key point was about the sample size calculation section. It is not clear what is the response level expected of the treatment arm and I was unable to replicate your sample size calculations. (see PDF)

6. PLOS authors have the option to publish the peer review history of their article (what does this mean?). If published, this will include your full peer review and any attached files.

**Do you want your identity to be public for this peer review?** For information about this choice, including consent withdrawal, please see our Privacy Policy.

Reviewer #1: No

Reviewer #2: No

---

## [Editor Report · Decision Letter 1]

13 Jul 2022

Collecting mortality data via mobile phone surveys: a non-inferiority randomized trial in Malawi

PGPH-D-22-00492R1

Dear Helleringer,

We are pleased to inform you that your manuscript 'Collecting mortality data via mobile phone surveys: a non-inferiority randomized trial in Malawi' has been provisionally accepted for publication in PLOS Global Public Health.

Best regards,

Rachel Hall-Clifford

Academic Editor